# Exploiting Potential Biotechnological Applications of Poly-γ-glutamic Acid Low Molecular Weight Fractions Obtained by Membrane-Based Ultra-Filtration

**DOI:** 10.3390/polym14061190

**Published:** 2022-03-16

**Authors:** Odile Francesca Restaino, Sondos Hejazi, Domenico Zannini, Concetta Valeria Lucia Giosafatto, Prospero Di Pierro, Elisabetta Cassese, Sergio D’ambrosio, Gabriella Santagata, Chiara Schiraldi, Raffaele Porta

**Affiliations:** 1Department of Experimental Medicine, Section of Biotechnology and Molecular Biology, University of Campania “Luigi Vanvitelli”, 80138 Naples, Italy; odilefrancesca.restaino@unicampania.it (O.F.R.); eli.cassese@gmail.com (E.C.); sergio.dambrosio@unicampania.it (S.D.); 2Department of Chemical Sciences, University of Naples “Federico II”, 80126 Naples, Italy; sondosmohammadhasan.hejazi@unina.it (S.H.); giosafat@unina.it (C.V.L.G.); 3Institute for Polymers, Composites and Biomaterials, National Council of Research, 80078 Pozzuoli, Italy; domenico.zannini@ipcb.cnr.it (D.Z.); santagata@ipcb.cnr.it (G.S.); 4Department of Agriculture, University of Naples “Federico II”, 80055 Naples, Italy; dipierro@unina.it

**Keywords:** poly-γ-glutamic acid, keratinocyte monolayers, oxidative stress, bio-based materials, hydrocolloid films

## Abstract

Since the potentialities of applications of low molecular weight poly-γ-glutamic acid (γ-PGA) chains have been so far only partially explored, the separation of diverse molecular families of them, as well as their characterization for potential bioactivity and ability to form films, were investigated. Two different approaches based on organic solvent precipitation or on ultra- and nano-filtration membrane-based purification of inexpensive commercial material were employed to obtain size-specific γ-PGA fractions, further characterized by size exclusion chromatography equipped with a triple detector array and by ultra-high-performance liquid chromatography to assess their average molecular weight and their concentration. The γ-PGA low molecular weight fractions, purified by ultra-filtration, have been shown both to counteract the desiccation and the oxidative stress of keratinocyte monolayers. In addition, they were exploited to prepare novel hydrocolloid films by both solvent casting and thermal compression, in the presence of different concentrations of glycerol used as plasticizer. These biomaterials were characterized for their hydrophilicity, thermal and mechanical properties. The hot compression led to the attainment of less resistant but more extensible films. However, in all cases, an increase in elongation at break as a function of the glycerol content was observed. Besides, the thermal analyses of hot compressed materials demonstrated that thermal stability was increased with higher γ-PGA distribution po-lymer fractions. The obtained biomaterials might be potentially useful for applications in cosmetics and as vehicle of active molecules in the pharmaceutical field.

## 1. Introduction

Poly-γ-glutamic acid (γ-PGA) is a poly-amino acid produced in nature by Gram-positive bacteria of the *Bacillus* genus (e.g., *Bacillus subtilis*, *B. subtilis subsp. natto*, *Bacillus licheniformis*), whose anionic polymeric structure is composed of D-glutamic acid and/or L-glutamic acid monomers linked by γ-amide bonds [1,2]. Nowadays, γ-PGA is considered a bio-homopolymer of high commercial interest, and it is already widely used in agriculture, environmental bioremediation, food industry, drug and cosmetic manufacturing, thanks to its biodegradable, edible, non-toxic, non-immunogenic and superabsorbent properties [2,3]. However, the different industrial applications of γ-PGA depend on its molecular weight (Mw), conformation, purity, solubility and pH conditions at which it is dissolved [2]. In water, at pH 7.0, γ-PGA solutions show high viscosity, whereas the homopolypeptide assumes an α-helix conformation in acidic conditions and a β-sheet-one at pH higher than 7.0 [4]. γ-PGA is also soluble in alcohols, but its solubility depends on the enantiomeric conformation. In fact, if the chains are composed of both D- and L-enantiomers, γ-PGA precipitates in ethanol, while if it contains only one type of enantiomer, it is soluble [3]. Furthermore, according to the microbial origin and to the culture conditions used in its fermentative production, the γ-PGA Mw ranges from less than 100 to 1000 kDa, and sometimes it can even reach 2000 kDa [1,5]. Ultra-high Mw γ-PGA chains (>1000–1500 kDa) can form colloids and could be used in the recycling of wastewater to remove heavy metals. Conversely, γ-PGA with a Mw > 700 kDa is widely used in cosmetics as skin moisturizing agent, being able to bind water in a proportion of 5000 times its weight, or in food industry as a thickener and cryoprotectant [1,5]. However, although both ultra-high and high Mw γ-PGA are quite exploited in industrial applications, their wider use is limited due to the high viscosity of their solutions at phy-siological pH values. Thus, in recent years, attention has been focused on the pre-paration of γ-PGA polymers of specific average sizes, mainly constituted from short Mw chains, that could be aimed at innovative commercial uses [2]. Oligomeric and low Mw (<100–200 kDa) γ-PGA have been used as drug carrier for tumor treatment [3] and, more recently, in the manufacturing of biodegradable films [6] potentially useful to produce materials with specific and tailored properties for applications in the pharmaceutical as well as cosmetic fields. Low Mw γ-PGA polymers are generally obtained by enzymatic digestion of those of high Mw chains of microbial origin, after their purification from the viscous fermentation broth.

The purification process has high costs and the recovery of the polymer can be hampered by the high viscosity of the fermentation broth [7,8]. The process generally includes centrifugation or sedimentation of the bacterial cells, γ-PGA recovery from the broth supernatant by alcohol precipitation (75–80% methanol or ethanol *v*/*v*) or addition of divalent copper salts. Ethanol precipitation did not assure the removal of proteins that precipitate along with PGA, thus further purification steps are required as concentration by ultra-filtration membranes, dialysis/diafiltration or chromatography to remove small molecules, such as salts, exhaust nutrients and even glutamate monomers [7,8,9,10]. Then, the purified product is dried by freeze drying or spray drying [7,8,9,10]. Recently, new strategies for the recovery and the purification of γ-PGA from the broth supernatant were reported that, for example, include four-unit operations, such as acidification, plate and frame filtration, cyclic ultra-filtration and freeze drying, in order to avoid any inefficient precipitation step and to reduce the purification costs [7]. In another case, a hybrid reactor system that integrated cross-flow microfiltration modules in a fermentation vessel was employed to continuously produce γ-PGA and directly purify it by using two-stage membrane treatments [10]. This approach assured continuous, high-yield production and productivity of γ-PGA by recycling of the bacterial cells and an efficient downstream process up to high product purity by removing any inhibiting fermentation products [10]. As the fermentative conditions influence the titer of γ-PGA production, as well as the Mw and polydispersity index of the homopolypeptide, and because the downstream purification process is a complex and not efficient procedure in which the different γ-PGA chain length populations are not separated, the commercial samples sold on the market often contain not pure, highly dispersed γ-PGA multiple forms with different Mw. As the Mw is critical for the γ-PGA applications, new downstream purification approaches are needed, as the classical precipitation procedures are not good in obtaining specific, highly homogenous Mw fractions, but only a mixture of different size populations. Therefore, the use of low Mw γ-PGA fractions for diverse biotechnological applications has been so far poorly investigated, particularly in the field of film manufacturing. So far, low Mw γ-PGA has been used to obtain films by formulating, casting and drying different film-forming solutions (FFSs) [6], while other methods, such as thermal compression of the polymer, have not been investigated yet.

In the present study, selected and highly homogenous γ-PGA fractions of low Mw were obtained on a large scale by a membrane-based purification process of ultra-filtration (UF), dia-filtration (DF) and nano-filtration (NF) from an inexpensive commercial source of γ-PGA (COM-PGA) and compared to a homopolypeptide preparation obtained from the same source by methanol precipitation (Figure 1). In order to quantify the γ-PGA concentration in all the samples, a new ultra-high-performance liquid chromatography (UHPLC) method was set up on the base of the L-glutamic acid monomer determination after acidic hydrolysis of the polymer. All the γ-PGA fractions were also characterized in terms of average Mw and poly-dispersity index by size exclusion chromatography coupled with a triple detector array (SEC-TDA) after determination, for the first time, of the γ-PGA dn·dc^−1^ value. Finally, the γ-PGA low Mw fractions obtained by UF, DF and NF and the homopolypeptide sample obtained by methanol precipitation (MET-PGA powder) were investigated for possible mammalian cell protection against dehydration and oxidative stress, as well as for the manufacturing of biomaterials potentially applicable in drug delivery systems and/or as dehydration masks in the cosmetic field. Manipulable γ-PGA films were produced not only by FFS casting but, for the first time, by hot compression of the powder samples and characterized in terms of mechanical, hydrophilic and thermal properties.

## 2. Materials and Methods

### 2.1. Materials

COM-PGA of microbial origin was from Xi’an Fengzu Biological Technology Co., Ltd. (Xi’an, China). According to the manufacturer, this industrial bulk contained not purified γ-PGA obtained by microbial fermentation and suitable only for agriculture uses. Glycerol (GLY) (99.5%) was supplied from Carlo Erba Company (Milano, Italy), as well as methanol (99.9%) All other chemicals used for the buffer of SEC-TDA and UHPLC analyses and the standards were of analytical grade and from Sigma-Aldrich (St. Louis, MO, USA). For the biological assays, the Dulbecco’s modified eagle medium (DMEM), the heat-inactivated fetal bovine serum (FBS), the penicillin and the streptomycin, as well as the phosphate-buffered saline (PBS) and the trypsin, were all provided by Gibco Invitrogen (Milan, Italy).

### 2.2. Poly-γ-Glutamic Acid Purification by Organic Solvent Precipitation

Absolute methanol (1.2 L) was added to COM-PGA powder (300 g). The mixture was stirred by using a refrigerated incubator shaker (ISF-1-W, Kuhner, Birsfelden, Switzerland) for 12 h at 4 °C and then centrifuged at 10,000 rpm for 10 min (Avanti J-20 XP, Beckman Coulter, Brea, CA, USA). The supernatant was discarded, while the pellet was dried at 30 °C to remove the excess of methanol (left panel of Figure 1). The obtained powder (MET-PGA) was kept at 4 °C and used for film preparation and characterization. 

### 2.3. Poly-γ-Glutamic Acid Fractionation by Ultra-Filtration and Nano-Filtration

COM-PGA (300 g) was dissolved overnight in 10 L of MilliQ water at room temperature under stirring conditions (600 rpm) and then microfiltered on a 0.65 μm polypropylene membrane (total filtering area of 0.05 m^2^, Sartopure PP2 MidiCaps, Sartorius Group, Göttingen, Germany) to remove undissolved materials by using a non-automatic Sartoflow Alpha system (Sartorius Group, Göttingen, Germany). As illustrated in the right part of Figure 1, γ-PGA fractionation was performed in three steps by using two different UF membranes in sequence and then by using a NF membrane. The microfiltered solution was first treated on 100 kDa cut-off polyethersulfone cassette membranes (total filtering area of 0.1 m^2^, Sartorius Group, Germany) and concentrated by using an automatic tangential flow filtration system (Uniflux 10, UNICORN, GE Healthcare, Chicago, IL, USA), equipped with a 10 L sample reservoir, feeding pumps, level sensors, pH and conductivity meters and connected to a software able both to control the process parameters and to collect data and process parameters continuously [11]. After that, 95.0% of the sample was filtered and a continuous DF was performed by adding two volumes of MilliQ water (with respect to the concentrated volume) to the 100 kDa retentate (R1). The 100 kDa permeate (P1) was then ultra-filtered on 3 kDa cut-off polyethersulfone cassette membranes (0.5 m^2^ of total filtering area, Sartorius Group, Germany) and concentrated by using a non-automatic Sartoflow Alpha system (Sartorius Group, Germany). The obtained 3 kDa retentate (R2) was then dia-filtered with two volumes of MilliQ water (with respect to the concentrated volume), while the 3 kDa permeate (P2) was then nano-filtered on 200 Da polyethersulfone spiral membranes (0.3 m^2^ of total filtering area, Fluxa Filtri, Milan, Italy) by using a non-automatic Sartoflow Alpha system (Sartorius Group, Germany). Finally, the nano-filtration retentate (R3) was dia-filtered with two volumes of MilliQ water (with respect to the concentrated volume) and concentrated, while the obtained nano-filtration permeate (P3) was discarded. During all the processes, data of the pressure and volumes were collected, while conductivity and pH of the solutions were measured. The trans-membrane pressure (TMP) was calculated by using the following formula:TMP = [(inlet pressure − retentate pressure)/2](1)

Fluxes were calculated as the volumes passed on a filtering area in 1 h (liter per square meter in 1 h, LMH). Samples were taken during each purification step to be freeze dried in order to determine their dry weight, the protein and inorganic material content and γ-PGA concentration. The selectivity of the membranes during the filtration processes was instead calculated on the base of the SEC-TDA analyses and on the dry weight of the samples according to the following formula:% Selectivity = [γ-PGA amount in the retentate (g)]/[γ-PGA amount in the retentate (g) + γ-PGA amount in the permeate (g)] × 100(2)

### 2.4. Analytical Methods

#### 2.4.1. Determination of the Dry Weight, Protein and Water Content of the Samples

Aliquots (10 mL) of samples of each step of the purification process, as well as the three collected retentate samples, were freeze dried (LIO 5P, 5Pascal, Milan, Italy) by using a previously reported method (18 h at −20 °C and 1.05 mbar and then 3 h at 20 °C and 0.04 mbar) [12] and weighted in order to determine their dry weight. The COM-PGA, the MET-PGA powder and the three retentate samples were also assayed to determine the percentage of total protein content with respect to the dry weight by using a colorimetric assay [13] and the bovine serum albumin (BSA) as standard (Bio-Rad Laboratories Inc., Hercules, CA, USA). The total protein percentage of the samples was calculated according to the following formula:% Protein/dry weight = % [(Protein sample(g)/dry weight sample (g)) × 100](3)

The water content of the COM-PGA, MET-PGA powder and of the three retentate samples was also investigated according to the method described by Shankar et al. [14].

#### 2.4.2. γ-PGA Concentration Analyses by Ultra-High-Performance Liquid Chromatography

γ-PGA concentration analyses were performed by ultra-high-performance liquid chromatography (UHPLC) after setting up a new method that exploited a previously reported analytical procedure for organic acid determination [12] and an acidic hydrolysis protocol previously used to depolymerize different polysaccharide chains and lignin–carbohydrate complexes [15,16,17]. The COM-PGA, the MET-PGA powder and all the other freeze-dried samples of the purification process were hydrolyzed with 5 M HCl, at a concentration of 2.5 g·L^−1^, for 6 h at 100 °C and 600 rpm (Thermomixer comfort, Eppendorf, Germany) to obtain L-glutamic acid monomers. Runs were performed by using an ultra-high-performance chromatography system (Ultimate 3000, Thermofisher-Dionex, Sunnyvale, CA, USA), by injecting 10 μL of the sample on a ion-exclusion column (Rezex ROA-organic acid H^+^, 300 × 78 mm, Phenomenex, Torrance, CA, USA) and by eluting in isocratic conditions with 0.1% H_2_SO_4_ for 25 min at 0.8 mL·min^−1^ and at 40 °C with detection at 200 nm. A calibration curve of the γ-PGA standard after hydrolysis was built in the range from 0.025 to 0.100 g·L^−1^ on the base of the L-glutamic acid monomer area. The peak area values of this calibration curve were compared with the ones of the L-glutamic acid standard curve, built after hydrolysis in the same range, and the percentage of the γ-PGA recovery after reaction was calculated. The values of the γ-PGA concentrations, obtained by UHPLC analyses, were multiplied for the sample volumes to calculate the total amount (g) of the polymer in each sample of the purification process. The percentage of γ-PGA recovery in each sample of the purification process, with respect to the COM-PGA content, was calculated according to the following formula:% γ-PGA recovery/COM-PGA sample = %[γ-PGA sample (g)/COM-PGA sample (g)] × 100(4)

The percentage of γ-PGA content in the COM-PGA, in the MET-PGA powder and in the three retentate samples with respect to their dry weight was calculated according to the following formula:% γ-PGA/dry weight = % (γ-PGA sample (g)/dry weight sample (g)] × 100(5)

The mass yield values were calculated as percentage ratio of the mass value of each sample on the mass value of the initial sample.

#### 2.4.3. γ-PGA Molecular Weight Analyses by Size Exclusion Chromatography with Triple Detector Array

Mw analyses of the COM-PGA sample, of the MET-PGA powder and of the three retentate samples were performed by using a SEC-TDA instrument (Viscotek, Malvern, Italy) equipped with a triple detector array, including a refractive index (RI) detector, a four-bridge viscosimeter (VIS) and two laser detectors of right-angle (RALS) and low-angle light scattering (LALS). Runs were performed by using two gel-permeation columns put in series (TSK-GEL GMPWXL, 7.8 × 30.0 cm, Tosoh Bioscience, Turin, Italy), equipped with a guard column, at pH 7.0 and at 40 °C and by eluting with 0.1 M NaNO_3_ at a flow rate of 0.6 mL·min^−1^, according to a previously reported method [6,18]. The instrument was calibrated by using a polyethylene oxide (PEO) standard (22 kDa PolyCAL, Viscotek, Malvern, Italy). All the retentate samples were analyzed in duplicate to determine the γ-PGA average Mw, the poly-dispersity index (Mw/Mn) and the intrinsic viscosity (IV) on the base of the detector signals and on the base of the γ-PGA dn∙dc^−1^ value by applying the equations reported by the manufacturer:RI signal = kRI · dn∙dc^−1^ · C(6a)
VIS signal = kVIS · IV · C(6b)
LALS signal = kLALS ·Mw ·(dn∙dc^−1^)^2^ · C(6c)
where IV is the intrinsic viscosity (dL·g^−1^); C is the concentration (mg·mL^−1^); dn∙dc^−1^ is the refractive index increment (mL·g^−1^); and kRI, kVIS and kLALS are instrumental constants obtained by the universal calibration with PEO, performed according to the manufacturer’s procedure (Viscotek, information available from http://www.viscotek.com, accessed date 10 March 2022).

The γ-PGA dn∙dc^−1^ value was never reported in the literature so far, and it was experimentally determined after injection in triplicate solutions of the commercial γ-PGA standard in a concentration range from 0.1 to 0.5 g·L^−1^, as previously described [18]. The commercial γ-PGA standard average Mw was 580 kDa ± 2 with a poly-dispersity index of 1.1 ± 0.1, as experimentally determined by using our instrument. By plotting the different γ-PGA concentrations versus the peak areas obtained from the RI signal, the γ-PGA dn∙dc^−1^ value was calculated according to the following formula:dn∙dc^−1^ = (Area RI/g·L^−1^·1000)/(kRI∙injection volume)∙RI of the solvent (7)
where the injection volume was 0.1 mL; the RI of the solvent was 1.33; the constant kRI was 9.72 × 10^7^. The representative of each peak was calculated as the percentage ratio bet-ween the single peak RI area divided by the sum of the RI areas of all the peaks in each chromatogram [18]. In the retentate samples, the percentages of representativity of the Mw populations were also determined by dividing the populations in four different ranges in order to calculate the selectivity of the membranes, as reported above: Fraction A—between 150 and 50 kDa, Fraction B—between 50 and 20 kDa, Fraction C—between 20 and 3 kDa, and Fraction D—lower than 3 kDa.

### 2.5. Biological Activity Assay

#### 2.5.1. Dehydration Assay

COM-PGA, MET-PGA powder and the three retentate samples were tested for their ability to counteract dehydration in a cell-based assay. The dehydration assay was performed by using a spontaneously transformed, non-tumorigenic human keratinocyte cell line (HaCaT) provided by Istituto Zooprofilattico (Brescia, Italy). The cells were cultured in DMEM medium supplemented with 10% (*v*/*v*) FBS, 100 U·mL^−1^ of penicillin and 100 μg·mL^−1^ of streptomycin, in 24-multiwell plates (5 × 104/well) until a 70% cell confluence was reached (157,000 ± 10 cells/well). At this point, the culture medium was removed, a wash with PBS was performed, and the keratinocyte cell monolayers were treated for 2 h with solutions of the COM-PGA sample, MET-PGA powder and of the three retentate samples at a concentration of 1.5 g·L^−1^ in DMEM. All experiments were performed in triplicate. The negative control (−CTR), instead, was carried out in triplicate simply by diluting the culture medium in PBS without any treatment with γ-PGA samples. After 2 h treatment, the samples and the negative control were dehydrated by removal of the medium and incubated at 37 °C until a stress response, in terms of morphological changes detected by using an optical microscope (100-fold magnification) (Axiovert, Zeiss, Oberkochen, Germany), was observed. The positive control (+CTR) of the experiment was constituted by cell mono layers not subjected to dehydration and kept in the presence of the medium during the whole experiment. Cell viability was determined by assessing cell metabolism through the NAD(P)H-dependent cellular oxidoreductase enzyme activity by a colorimetric assay. Cells were incubated for 3 h with the 3-(4,5-dimethylthiazol-2-yl)-2,5-diphenyltetrazolium bromide (MTT) reagent (Cell proliferation Kit I, MTT, Merck, Rome, Italy) at 37 °C and then treated with 0.1 M HCl in isopropanol (Sigma-Aldrich, St. Louis, MO, USA) for 15 min at room temperature. The NAD(P)H oxidoreductase activity was evaluated by measuring the absorbance at 570 nm (PA800, Beckman Coulter, Brea, CA, USA), and the percentage cell viability in the γ-PGA treated samples was compared to the cell viability in the negative control.

#### 2.5.2. Oxidative Stress Assay

COM-PGA, MET-PGA powder and the three retentate samples were compared for their ability to counteract the H_2_O_2_ oxidative stress in a cell-mediated test (HaCaT) as rescuing agents. In the experiments 150,000 cell/well were seeded in 24-multiwell plates and cultivated for 24 h until a 40% cell confluence was reached (90,000 ± 10 cells/well). Then, the medium was removed, a wash with PBS was performed and 50 µM H_2_O_2_ (Sigma-Aldrich, USA) was added. After 30 min, the samples were observed by using the optical microscope in order to evaluate the eventual morphological changes due to the oxidative stress. Then, the H_2_O_2_ solution was removed, a wash with PBS was performed, and the keratinocyte mono layer cells were treated for 24 h in triplicate with solutions of COM-PGA, MET-PGA powder and of the three retentate samples at a concentration of 1.5 g·L^−1^ in DMEM (except for the CTR samples). The MTT viability assay was performed after 24 h, as previously described, and the percentage of cell viability of the samples was compared with the negative control. The negative control was simply prepared by diluting the culture medium in PBS and then performing the oxidative stress, while the positive control was not subjected to treatment with H_2_O_2_, and the cells were simply kept in the presence of the medium during the whole experiment.

### 2.6. Film Preparation and Characterization

#### 2.6.1. Preparation of Films by Casting

Different FFSs (30 mL) containing 800 mg of either MET-PGA powder or retentates containing different γ-PGA fractions obtained by UF (R1, 55 kDa, R2, 18 kDa and R3, 6 kDa), were prepared by adjusting the pH to different values (2.0, 2.5, 3.0, 3.5, 4.0) by addition of 2.0 N HCl. All the FFSs were ultrasonicated (Bandelin SONOPULS ultrasonic homogenizers, Binder, Tuttlingen, Germany) for 20 min at 85% power by using 2 × 10% cycle. The FFSs were prepared in the absence or presence of different concentrations of GLY (1–10%, *w*/*w* γ-PGA). FFSs were poured into 60 × 15 mm polystyrene Petri dishes and left to dry in climatic chamber (25 °C and 45% relative humidity (RH) for 48 h). Dried films were conditioned at 25 °C and 50% RH for 2 h before the peeling and characterization.

#### 2.6.2. Preparation of Films by Thermal Compression

Films of quite uniform thickness (≈500 μm) were prepared by compression molding in a Carver lab press (Carver Inc., No. 4122-12-12H, Wabash, IN, USA). Prior to film production, 2 g of γ-PGA powder samples (COM-PGA, MET-PGA, R1 and R2) were homogeneously mixed with 5% or 10% GLY (*w*/*w* of γ-PGA) in a mortar by pestle. Each single dough obtained was finally shaped into small spherical pellets, then subjected to thermal compression. The samples were sandwiched between two flats of polytetrafluo-roethylene foils and heated between the plates of the press up to 150 °C for 2 min at 500 PSI. Then, the samples were removed from the hot press and cooled down to room temperature before the film peeling. Films were always conditioned at 25 °C and 50% RH for 2 h before their characterization.

#### 2.6.3. Film Moisture Content and Solubility

For the determination of the moisture content (MC) of the films, small pieces of them (≈2 × 2 cm) were weighted, put in aluminum dishes and dried in an oven at 105 °C for 24 h. The weight before and after drying was recorded to calculate the MC by the following equation:% MC = [(Wi − Wd)/Wi] × 100(8)
where Wi and Wd represent the weights of the initial and dried film, respectively.

Water solubility (WS) of the films was instead calculated by determining the initial weight (Wi) of the film samples after oven drying at 105 °C for 24 h and by immersing the dried specimens in 50 mL of distilled water continuously shaken for 24 h at 25 °C. Then, the insoluble film pieces were oven dried again at 105 °C to obtain the final dry weight (Wf) [19]. Film WS (%) was calculated according to equation
% WS = [Wi × 100]/Wf(9)

#### 2.6.4. Film Mechanical Properties

All handleable films were cut into 1 cm × 8 cm strips and analyzed for their mechanical properties (tensile strength, TS; elongation at break, EB; Young’s module, YM) by means of a dynamometer Instron (Engineering Corp., Norwood, MA, USA), according to the procedure described in ASTM D882–10 [20]. Five specimens of each film sample were analyzed (5 cm gage length, 1 kN load and 20 mm/min speed) taking into consideration film thickness that was measured in five different points with a micrometer with a sensitivity of 0.001 mm (Electronic digital micrometer, DC-516, Alpa, Albiate Monza e Brianza, Italy).

#### 2.6.5. Film Thermal Properties

Thermogravimetric analysis of MET-PGA-, R1- and R2-derived films was performed using a thermogravimetric analyzer TGA/DTG Perkin-Elmer PyrisDiamond, equipped with gas station. Samples (3–4 mg) were placed in an open ceramic crucible and heated from 25 °C to 600 °C at a speed rate of 10 °C·min^−1^, under nitrogen flow of 30 mL·min^−1^. Before testing, samples were conditioned for 24 h at 25 °C and 50% RH. Thermal properties of MET-PGA-, R1- and R2-derived films were investigated by using a Q2000 T zero differential scanning calorimeter (DSC), TA Instrument (New Castle, DE, USA), equipped with a liquid nitrogen accessory for fast cooling. The calorimeter was calibrated in temperature and energy using indium. Dry nitrogen was used as purge gas at a rate of 30 mL·min^−1^. Samples (3–4 mg) were weighed and placed individually in aluminum pans; an empty pan was then used as reference. DSC measurements were performed in double heating run; the first one, occurring from 50 to 200 °C, at 10 C·min^−1^, could reproduce the thermal history of the samples. After an isothermal step of one min, non-isothermal crystallization experiments were performed by cooling the specimens up to −50 °C, at a rate of 10 C·min^−1^. Finally, a second heating ramp from −50 °C to 250 °C at 10 °C·min^−1^ was recorded. Before testing, the samples were conditioned for 24 h at 25 °C and 50% RH.

### 2.7. Statistical Analysis

For cell viability analyses, data were reported as averaged values of three replicates with their standard deviation values and were statistically analyzed by using Excel program (Microsoft, Redmond, WA, USA), using the Student’s *t*-test, considering statistically significant differences between data with *p* < 0.05. In terms of film characterization, in order to determine a significant difference between treatments, one-way analysis of variance (ANOVA) and Duncan’s multiple range tests (*p* < 0.05) were performed using the Statistical Package for the Social Sciences (SPSS19, SPSS Inc., Chicago, IL, USA) software.

## 3. Results

### 3.1. γ-PGA Quantitative Determination by UHPLC and Purification by Organic Solvent Precipitation

In order to determine the γ-PGA concentration in the commercial material and in all the purified samples, a new UHPLC method was set up on the base of the L-glutamic acid monomer determination after acidic hydrolysis of the homopolypeptide. The L-glutamic acid standard shows a peak at 5.5 min, as well as the γ-PGA standard after acidic hydrolysis (92.5% of the γ-PGA standard was recovered after reaction). Therefore, a calibration curve was built in the linearity range from 0.025 to 0.100 g·L^−1^ (Appendix A). The obtained results indicate that the commercial dry sample contained only 51.5% of γ-PGA. Thus, 300 g of COM-PGA, containing around 154 g of γ-PGA, was mixed with methanol, and the obtained precipitated material (250 g) was dried overnight in the oven at 37 °C to eliminate methanol. Afterward, the dry material was grinded to obtain a powder (MET-PGA powder) that was further characterized and used for film preparation. At the end of this treatment, only 83% of the original powder was recovered (250 g) with a consequent γ-PGA content of 128.2 g. A similar percentage of recovery was previously reported by Manocha and Margaritis [21] by using ethanol to precipitate γ-PGA. The conductivity and the pH of the MET-PGA powder, when dissolved in water, were 13 mS·cm^−1^ and pH 6.4, respectively.

### 3.2. γ-PGA Purification by Using Filtration Membranes and Characterization

The COM-PGA was purified on different cut-off membranes, as illustrated in the right panel of Figure 1, with the aim to recover fractions with the highest polymer concentration and of homogeneous averaged Mw size. The entire process is summarized in Table 1 and Figure 2. An amount of 300 g of the COM-PGA was first solubilized in water (10 L), but 9.2% of the sample remained insoluble.

Thus, a micro-filtration step was performed to remove other insoluble material contained in the starting sample. A total of 85.7% of the initial γ-PGA was recovered in the microfiltered permeate that was initially concentrated and dia-filtered on 100 kDa membranes. The process parameters are reported in Figure 2A.

The 100 kDa retentate (R1) had a concentration factor of 13.3, and it contained 15.5% of the initial material. The resulting permeate sample (P1) was then concentrated and dia-filtered on 3 kDa membranes (Figure 2B), achieving a conductivity decrease from 8.8 to 5.8 mS·cm^−1^. The corresponding retentate (R2) was 11.6-fold concentrated, and it contained 12.3% of the initial COM-PGA. The resulting permeate (P2) was then treated on the nano-filtration membranes (Figure 2C), and during the process, conductivity decreased up to 4.0 mS∙cm^−1^. The high TMP values reached did not permit a further dia-filtration step. The nano-filtration process allowed a recovery of 50.4% of the initial γ-PGA in the retentate (R3) (concentration factor of 4.4), while the remaining 3.3% was found in the nano-filtration permeate (P3). The selectivity values were 18.8%, 16.6%, 93.9% for the three steps of purification, respectively. In conclusion, the total γ-PGA recovery of the whole membrane process in the three retentate samples was 77.3%; the three retentate fractions were freeze dried, and their γ-PGA content resulted in an increase compared to the COM-PGA, since the γ-PGA content on the dry weight, also considering a water percentage of the freeze-dried samples between 6.0 and 10.0%, was 69.5%, 58.6% and 64.3%, respectively. The percentage protein concentrations on the recovered freeze-dried material of the three retentate samples was very low, between 0.3 and 3.2%.

### 3.3. γ-PGA Molecular Weight Analysis by SEC-TDA

To perform the SEC-TDA analyses of the samples, the dn∙dc^−1^ value of the γ-PGA standard was experimentally determined for the first time, and it emerged as 0.183 mL·g^−1^ (Appendix A). COM-PGA, MET-PGA and the three obtained retentates were analyzed (Figure 3A). In Appendix A, their complete chromatograms showed the diverse bio-polymer families that corresponded to the average Mw values reported and highlighted in Figure 3B. Analyses showed the presence of three different populations of γ-PGA with low and ultra-low averaged Mw(between 78.3 and 5.6 kDa, with a poly-dispersity index between 1.07 and 2.73). The lowest Mw (5.6 kDa) fraction of γ-PGA constituted the main peak of the commercial sample with a representativity of 64.5%. The precipitation of the COM-PGA by methanol roughly halved the ultra-low Mw population, since the SEC-TDA analysis showed the presence of three peaks, with averaged Mw values like the values of the commercial sample (between 64.0 and 3.0 kDa) but with different representativity. Conversely, the representativity of the other two populations with higher Mw (64 and 20.6 kDa) increased by about 1.6- and 1.9-fold. The membrane-based purification process allowed for obtaining γ-PGA fraction samples that were more homogeneous in size. More specifically, R1 contained a homopolypeptide fraction with an average Mw of 54.7 kDa, a poly-dispersity index of 1.56 and a representativity of 98.5%. In R2, the main peak of 17.9 kDa with a representativity of 87.4% was found (Figure 3), together with a higher Mw peak of 52.7 kDa, accounting for the 12.6% of γ-PGA. Finally, γ-PGA chains with Mw values lower than 10 kDa were collected only in R3. It is worthy to note that γ-PGA of average 3000 kDa Mw (between 2761 and 3254 kDa) was found in very small percentages (up to 1.5% of representativity) in all the analyzed samples.

### 3.4. Biological Activity: Dehydration and Oxidative Stress Assays

The biological activity of COM-PGA, MET-PGA powder and of the γ-PGA-containing R1, R2 and R3 fractions was investigated using HaCaT cells by an in vitro model for challenging both dehydration and oxidative stress. The dehydration test showed an 80% decrease in viability of the negative control, and a similar effect was observed in the presence of the COM-PGA sample (Figure 4A). Conversely, the addition of all the other γ-PGA samples was able to restore cell viability to various extents. In particular, R1 and R2 γ-PGA-containing fractions caused a marked improvement, with a recovery higher than 2.5-fold, whereas cell viability increased by less than 30% and 40% by employing R3 and MET-PGA powder (normalized to the positive control), respectively (Figure 4A).

As far as the oxidative stress applied to HaCat monolayers is concerned, it reduced cell viability by about 70% compared to the positive control, while all γ-PGA-containing fractions were found to determine a significant rescue of the cell metabolism when added after the oxygen peroxide treatment (Figure 4B). In fact, while the COM-PGA seemed unable to improve cell viability, both MET-PGA powder and all the γ-PGA fractions obtained by ultra-filtration had a marked enhancing effect and, among them, the R3 fraction showed by far the best performance.

### 3.5. γ-PGA Film Preparation by FFS Casting/Drying and by Hot Compression

γ-PGA-based films were prepared by two different procedures, either by casting and drying of the FFSs containing 800 mg of the homopolypeptide and GLY at two different concentrations (5 and 10%, *w*/*w* of γ-PGA) or by hot compression for 2 min at 150 °C and 500 p.s.i. of the powders (2 g) of the different γ-PGA fractions, obtained by methanol precipitation or by ultra-filtration, after their mixing with GLY (5% *w*/*w* of γ-PGA). As it is possible to see from panel A of Figure 5, all the films obtained by the FFS casting appeared brownish, and only the MET-PGA-based FFS containing 10% GLY and all the R3-based FFSs, prepared both in the absence or presence of GLY [22], did not allow for obtaining any manipulable films after casting and drying.

In contrast, all the films prepared with the R1 γ-PGA fraction (55 kDa), both in the absence and in the presence of the two different GLY concentrations, were easy to be peeled off from the plates and were very manipulable, even though they presented some spots on their surface. Finally, the R2 γ-PGA fraction (18 kDa) gave rise to brittle films when prepared either in the absence or presence of 5% GLY; conversely, brown and transparent films without any irregularities were obtained in the presence of 10% of GLY.

Panel B of Figure 5 shows the films obtained by hot compression of the γ-PGA powders (COM-PGA, MET-PGA, R1 and R2 γ-PGA fractions) previously mixed with 5% GLY, as described in Materials and Methods. These films showed a dark brown color and were handleable and quite homogenous, whereas no manipulable films were obtained by hot compression of the R3 γ-PGA powder, similarly to what occurred after casting and drying of the FFS containing R3 γ-PGA fraction.

### 3.6. Film Characterization for Mechanical and Hydrophylicity Properties

All the films prepared with the two different procedures were analyzed according to their mechanical (TS, EB and YM) and hydrophilicity (MC and WS) properties, and the detected results, reported in Table 2, were compared. It was possible to analyze only the peelable and well-shaped materials, since unmanipulable films (either * sticky or ** brittle) were obtained by casting and drying the FFSs containing COM-PGA, with or without GLY, or MET-PGA and 10% GLY (both originating sticky materials) and R2 γ-PGA fraction prepared in the absence or presence of 5% GLY (originating brittle materials). Moreover, similar unmanipulable materials were obtained in the attempt to prepare γ-PGA films by hot compression from COM-PGA and 10% GLY or from MET-PGA in the absence of GLY, which gave rise to sticky and brittle materials, respectively. Finally, not even R3 γ-PGA fraction was able to produce manipulable films, both in the absence or presence of GLY, neither by casting and drying nor by hot compression of its powder. It is worthy to note that the thickness values of the films obtained by hot compression were markedly higher (2–4 times) as a consequence of the higher amount of the homopolypeptide used (2 g with respect to 0.8 g occurring in the cast FFSs) and that the effect of the presence of GLY in increasing film thickness was observed only when the films were prepared by casting and drying the γ-PGA FFSs.

The analyses of the mechanical properties showed a marked plasticizing effect of GLY in the films prepared by both procedures with all the different γ-PGA samples, as TS and YM always decreased, and EB values concurrently increased, as a function of GLY concentration enhancement. In particular, it deserves to be highlighted that both R1 and R2 γ-PGA films prepared in the presence of GLY (especially R2-derived film with 10% GLY) were endowed with a remarkably high extensibility. Conversely, the films obtained by hot compression of all the γ-PGA powder samples exhibited very low TS values (<1.0 MPa) and, as a consequence, undetectable YM values. Thus, it seems that the hot compression procedure led to production of less resistant but much more extensible films.

The films were also analyzed for their MC and WS after immersing the samples in distilled water for 24 h. As shown in Table 2, the materials seemed to behave in a very similar way regardless of the method used for their manufacture. In fact, all the films showed a WS higher than 80%, except for R1-based films prepared by means of thermal compression in the absence or presence of 5% GLY, the WS of which was 62 and 71%, respectively. However, the hot compressed materials were found to possess a lower amount of water, as the measured MC was lower than that of the respective cast films prepared under the same conditions, with the only exception of the R2 γ-PGA films manufactured with 10% GLY.

### 3.7. Thermal Properties of γ-PGA Different Samples and of the Derived Films Manufactured by Hot Compression

The thermal properties of both MET-PGA and R1 and R2 powders, as well as of the derived films manufactured in the absence or presence of 10% GLY by hot compression, were evaluated by both thermogravimetric analysis and differential scanning calorimetry (DSC). However, it is worth considering that, in general, γ-PGA could exhibit different thermal characteristics depending on the microorganism used for fermentation and the fermentation conditions, which can in turn influence the Mw and, consequently, the physicochemical properties of the analyzed homopolypeptide sample [23].

#### 3.7.1. Thermogravimetric Analysis

The thermogravimetric (TGA) and the differential thermogravimetric (DTG) curves of MET-PGA, R1 and R2 γ-PGA films, without and with glycerol, are reported in Figure 6A,B and Figure 6D,E, respectively. All curves were normalized with respect to the starting sample size.

In order not to overburden the paper content, finalized to mainly emphasize the role of films, all the thermal analysis of powder-based samples, together with their interpretation and discussion, were postponed to Appendix A). In Figure 6A,B,D,E, the thermograms of neat and GLY-doped films are reported. Generally, from the analysis of weight loss curves (Figure 6A,D), different mass evolution steps could be detected, highlighting the complex pattern of film thermal degradation profiles. This outcome was better highlighted by analyzing the DTG curves of all the films. Specifically, for all neat samples (Figure 6B), at least three different regions of water evolutions could be found with their corresponding maxima peaks at around 50, 100 and 150 °C. In fact, as widely reported in the literature, the investigated region can be ascribed to free, bound and strongly bound water release, mostly if hydrophilic and hygroscopic polymers are concerned. It should be considered that γ-PGA-based samples are characterized by the presence of many polar groups, such as amino, carboxyl and hydroxyl residues, mostly involved in hydrogen bonding with water and, in any case, able to absorb and/or entrap water molecules in their three-dimensional networks. These findings, likewise well documented in the literature by some of the authors of the current paper [6], could also be explained by considering the thermal processing undergone by the films (compression molding), during which strong hydrogen bonds between water molecules and polymer polar residues occurred, responsible for water entrapment inside the polymer matrix network, releasable only at high temperature. Similar results were also found in the presence of GLY (Figure 6D,E); in these samples, the presence of two main regions of water evolution is well evident. The first one, up to 100 °C, was likely associated with the free and weakly bound water release, whereas the second temperature range between 100 and 200 °C was likely due to water evaporation coming from GLY-rich region, in which hydrogen bonding occurred. In fact, it is worthy to mention that water released at high temperature also came from the condensation reactions following polymer thermal degradation. Sabbah et al. [6] found that γ-PGA underwent the first depolymerization process at around 200 °C, consisting of an end-of-chain unzipping cyclodepolymerization reaction, leading to pyroglutamic acid. The short chains formed upon heating could depolymerize at around 250 °C, as shown by the small shoulder peak in both neat and doped films, whereas the main degradation process of high molecular weight polymers occurred in the wide range of 280–500 °C. From the comparison of DTG thermograms reported in Figure 6B,E, it is worthy to note that in the presence of GLY, a clear splitting of the different γ-PGA Mw fractions degradation rate occurred [24]. Specifically, as shown in Table 3 and Figure 6E, the lower molecular weight fractions, likely involved in hydrogen bonding with GLY, showed a maximum degradation rate peak at around 280 °C, about 20 °C lower with the analogous chain fractions of the neat polymers, degrading at 300 °C (Figure 6B) [25]. This result was ascribed to the increase in free volume and macromolecular mobility of the plasticized systems, as widely reported in the literature [26]. It was also likely that the thermal depolymerization of lower Mw macromolecular chains occurred simultaneously with GLYdegradation, thus explaining the higher peak intensity of this region. The third thermal peak, related to the higher γ-PGA molecular weight decomposition, occurred at 330 °C in all the samples, thus confirming both that the effect of the plasticizer was mainly marked in smaller Mw γ-PGA polymer residues and that a phase separation occurred during hot compression molding between regions of different molecular weights [27,28]. The final step of decomposition, observed by ana-lyzing the film samples in 450–550 °C range, was likely associated with the decarboxylation process leading to unsaturated chain fragments, non-volatile compounds in inert atmosphere (nitrogen).

#### 3.7.2. Differential Scanning Calorimetry

DSC thermograms, related to the second heating ramp of MET-PGA, R1 and R2 films, are reported in Figure 6C,F. Apart from the expected chemical reaction, the first thermal heating of polymers by DSC analysis erases their thermal history (Appendix A), while the second thermal run accounts for the real polymer properties at mole-cular level, evidencing all the involved transition phenomena [29]. In fact, due to the instrument limiting upper value of temperature, it was not possible to take a suitable guess related to the starting endothermic peaks visible for all samples at around 250 °C. In any event, from the differential thermal analysis (DTA) (Appendix A), detectable by the same thermogravimetric analyzer used for TGA/DTG analysis in the range of 25–600 °C, and able to provide qualitative information on liable polymer transitions, no peaks could be found, apart from the ones related to polymer thermal degradation, better highlighted by DTG thermograms, as previously discussed. Thus, from DSC profiles, the starting endothermic peak found at 250 °C could likely be due to polymer degradation events. Nevertheless, from DSC analysis of the second heating run, it was possible to detect the glass transition temperature (Tg) of all the samples. The results, reported in Table 3, demonstrated the plasticizing action of GLY, as the Tg decreased for all the tested films [30].

## 4. Discussion

Up to now, the use of different low Mw γ-PGA chains as natural and biodegradable sources for diverse biotechnological applications has been poorly investigated [6]. In particular, little exists in the literature on the specific application of low Mw fractions of the homopolypeptide as cosmeceutical agent or to produce biodegradable films. As the γ-PGA functions change according to the size, their homogeneous low Mw fractions might potentially exhibit different properties compared to both high or very high Mw γ-PGA or to the mixtures of different Mw chains of the homopolypeptide. Thus, a key point in assessing specific polymer properties and potential functions is the necessity to find new protocols to purify and prepare γ-PGA size-specific fractions and to characte-rize them in terms of concentration and Mw [2]. So far, the γ-PGA quantitative determination has been performed by spectrophotometric assays (e.g., colorimetric test of safradin-induced precipitation, or after cetyltrimethylammonium bromide precipitation, or UV absorbance) [31,32]. Although these methods are apparently rapid, most of them require extensive purification of the samples before the analyses to avoid interference of other components in the estimation of the real titer. Zeng et al. [32] used an HPLC method for the γ-PGA determination after a strong hydrolysis (i.e., 105 °C for 8 h). In the present study, a new UHPLC method was set up to precisely quantify γ-PGA, occurring even in raw samples, on the base of the L-glutamic acid amount after milder acidic hydrolysis of the samples and ion-exchange chromatography separation. This hydrolysis procedure, which already proved to be effective for the quantitative determination of other macromolecules [16,17,18], emerged as highly reliable (92% of recovery) also in evaluating the amount of γ-PGA. Besides, so far, γ-PGA molecular weight analyses have been performed by using methods such as SDS-PAGE (with methylene blue or alcian blue staining, as well as with densitometry band detection) and gel-permeation chromatography with RI detection [32]. In the present study, instead, we used SEC-TDA analyses that also involve the contemporary detection by light scattering signals and intrinsic viscosity, thus obtaining a more reliable determination of the γ-PGA Mw and poly-dispersity index. In fact, improving our previous results [6], the γ-PGA dn·dc^−1^ value was also experimentally determined for the first time, since in the literature, only the increment of the calculated, and not experimental, value of L-glutamic acid is reported [33]. Furthermore, improved procedures for γ-PGA purification are needed for a better characterization of the homopolypeptide. Alcohol precipitation is one of the main methods reported. However, it is not specific, and undesired polysaccharide and protein impurities contained in the fermentation broth generally remain in the γ-PGA samples and must then be removed with following acid or protease treatments [31]. Moreover, this method involves the consumption of high volumes of solvent, which raises environmental concerns, especially in large-scale production of the homopolypeptide [10]. Finally, alcohol precipitation does not fractionize γ-PGA in homogeneous Mw samples. In this respect, various methods, such as alkaline or enzymatic hydrolysis and ultrasonic irradiation, have been used to degrade the ultra-high and high Mw γ-PGA, obtained by fermentation, into low Mw homopolypeptide chains [31]. However, even these methods have limits, by degrading the polymer producing highly poly-dispersed molecular species. Size-specific γ-PGA samples (between 13 and 1300 kDa) have been previously obtained by using a hydrophilic silica-based anion exchange [31] or by gel-permeation chromatography [34]. However, these kinds of chromatographic methods are not applicable for large-scale γ-PGA preparations [34]. Therefore, membrane-based purification appeared more selective than alcohol-based precipitation and easy to scale up [11]. In the literature, ultra-filtration with polyethersulfone hollow fiber membranes (cut-off of 500 kDa, 300 kDa and 30 kDa) have been previously used to concentrate γ-PGA from a highly viscous culture broth and to reduce the amount of alcohol to be used in a second step of recovery [10]. An ultra-filtration process (with plate and frame cassettes of 10 and 100 kDa), coupled with nano-filtration was instead used in a previous work on lab scale to purify a small amount of γ-PGA (25 g). In this paper, we attempted a scale up of the previously developed tangential flow-based ultra-filtration process [6], to better recover some of the fractions with slight modifications of ultra-filtration membrane size in the second step, in order to purify a larger amount of COM-PGA (300 g) that contained a mixture of three different low Mw γ-PGA populations. The fractions were then compared with the sample obtained from the same raw material by methanol precipitation. This last procedure partially removed the very low Mw chains (<6.0 kDa) of the homopolypeptide from the COM-PGA, thus increasing the representativity of the other two macromolecular species and delivering high γ-PGA recovery percentages, but without an extensive purification of the commercial sample or selecting a precise size population of the homopolypeptide. The membrane-based process, instead, allowed for obtaining fractions with quite homogeneous sizes (54.7 and 17.9 kDa) and a higher purity grade. These fractions showed biological properties in terms of rescue effect against dehydration and oxidative stress. In fact, COM-PGA was not able to preserve the viability of the HaCaT cell monolayers, whereas the γ-PGA fractions with specific Mw sizes, such as R1 (54.7 kDa) and R2 (18.0 kDa), protected the cells from the dehydration stress and were rescuing the detrimental effect of oxygen peroxide treatment, with a significant outcome also by the R3 (6 KDa) γ-PGA fraction. The homopolypeptide size was shown to also affect the properties of the homopolymer-derived biomaterials. It is inte-resting to note that R3, containing very low Mw species of γ-PGA, was not able to produce any kind of films, neither in the absence nor presence of plasticizer, while COM-PGA, MET-PGA and the two retentate samples, R1 and R2, allowed for obtaining, under selected experimental conditions (i.e., pH and GLY concentration), manipulable films by casting/drying of FFSs or by hot compression of the sample powders. The mechanical properties of the manufactured γ-PGA-based materials demonstrated that GLY was able to promote a marked extensibility of these films, reducing contextually their TS and YM. These findings suggest a potential cosmetic application aimed at the preparation of hydration masks where the ability to stretch the skin is a prerequisite of paramount importance. In particular, an extraordinary extensibility (EB value of 785%) of the R2-based films when manufactured by hot compression and plasticized with 10% GLY was demonstrated. This behavior might be due to the fact that the thermal pressing at 150 °C of R2 powder mixed with 10% GLY led to an extensive esterification reaction between the carboxyl groups of the low Mw species of γ-PGA occurring in R2 sample and the hydroxyl groups of GLY that, in turn, improved the extensibility of the films as a consequence of the formation of a high number of covalent stable bonds between the plasticizer and the polymer matrix [35].

Finally, the thermal characterization of the materials obtained by hot compression showed an increase in their thermal stability corresponding to the enhancement of the Mw of γ-PGA molecular species. The thermogravimetric analyses of the different γ-PGA samples indicated at around 330 °C the peak of the thermal decomposition of the higher Mw species of γ-PGA (54.7 kDa) contained in R1 fraction, whereas the peak corresponding to the decomposition of the lower Mw species of γ-PGA (17.9 kDa) present in the R2 samples occurred just below 300 °C. In addition, from DSC analyses, it was possible to assess that R2 sample evidenced the lower glass transition temperature, as expected, given its lower molecular weight. Moreover, a worthy plasticizing effect of GLY was extended to all the films, thus confirming, at molecular level, the previous improvement of mechanical properties for all the GLY-doped samples. A recent study of Lee et al. [35] on the design of films from cellulose graft copolymers obtained by carver press demonstrated that the hot compression really led to an evident thermos-plasticizing effect. In conclusion, fractionated γ-PGA may be used for film formation with diverse possible uses, among which the bioactivity data here reported suggest some cosmeceutical applications. It is worth to mention that, for the first time, γ-PGA was used to prepare biomaterials by both casting and hot compression method. Although casting is the most successful technique for preparing homogeneous materials, it is less suitable for a large-scale process. In fact, the hot compression requires less time to produce materials and, therefore, it has been demonstrated to be a promising method for packaging ma-nufacture. Besides, the thermally obtained films ensure a superior processability and may also be proposed for packaging purposes. In the future, emphasis should be given to scale up experimentation on this hot compression method for the commercialization of γ-PGA-based biomaterials. However, further attempts are necessary in order to increase the very low TS found for the films prepared by means of this method (Table 2).

## 5. Conclusions

The potential of γ-PGA as raw ingredient for ready-to-use market products has been extensively explored so far, even though the majority of the commercially inexpensive preparations of the homopolypeptide seem to result in low quality and purity. To better highlight the potential use of specific homogenous low molecular size populations of γ-PGA obtained by fermentation, a specific three-step membrane process was developed up to 300 g scale of the commercial product and compared to γ-PGA obtained from the same original source after precipitation by methanol. The retentate fractions, also cha-racterized by new analytical methods, permitted the obtaining of films by both casting and hot compression with good technological properties and diverse potential applications. Furthermore, since the results obtained by two skin damage experimental models showed that the low Mw forms of the homopolypeptide were able to counteract the desiccation and the oxidative stress of keratinocyte monolayers, all these data suggest possible applications of the γ-PGA-based films also as protective co-smeceutical devices.

## Figures and Tables

**Figure 1 polymers-14-01190-f001:**
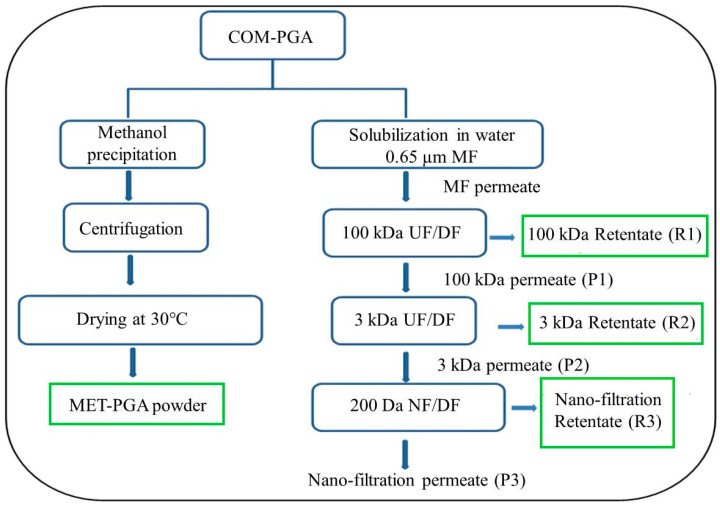
Flowchart of the processes for obtaining partially purified γ-PGA from COM-PGA by methanol precipitation (MET-PGA powder) or low Mw fractions of the homopolypeptide by membrane-based purification by micro-filtration (MF), ultra-(UF)/dia-filtration (DF) and nano-filtration (NF). The main γ-PGA fractions (MET-PGA powder, R1, R2 and R3), here reported in the green boxes, were then characterized and assayed for their properties.

**Figure 2 polymers-14-01190-f002:**
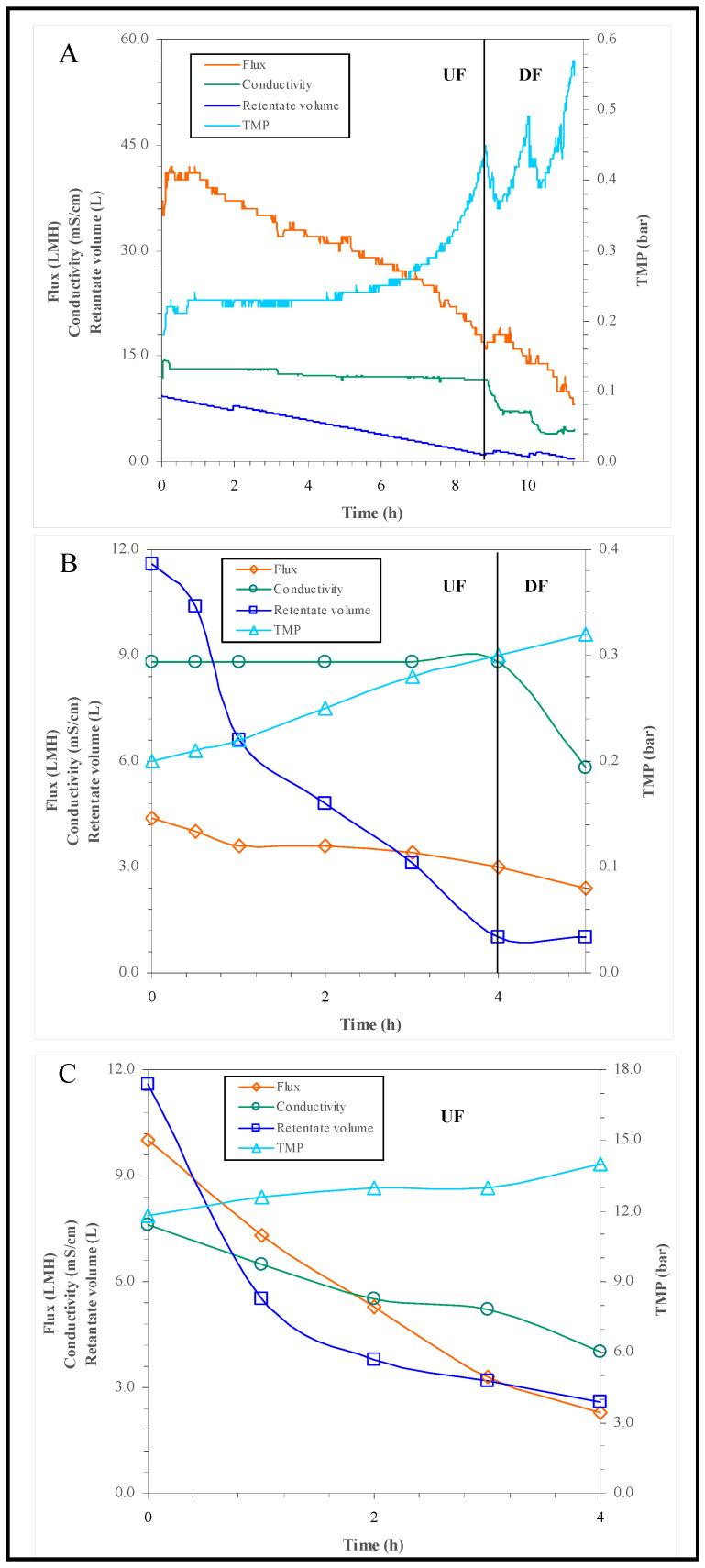
γ-PGA ultra-filtration (UF) and dia-filtration (DF) phases of 100 kDa (**A**), 3 kDa (**B**) membrane processes and the ultra-filtration phase of the nano-filtration purification (**C**), as indicated in the graphs. The curves of TMP, flux, conductivity and retentate volume during the processes are reported.

**Figure 3 polymers-14-01190-f003:**
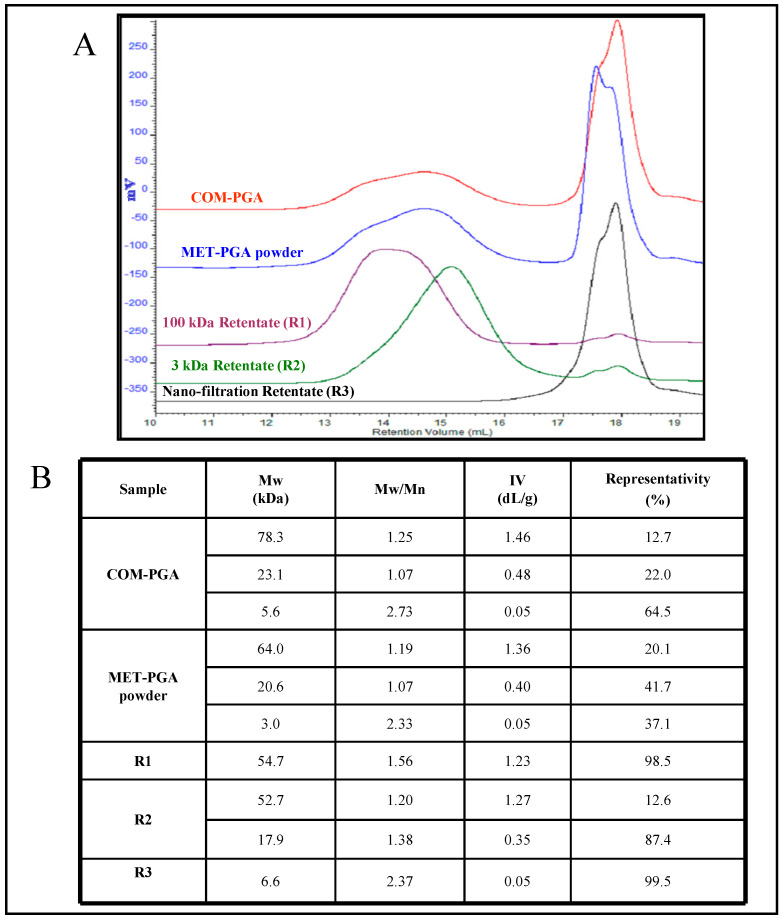
Overlaid SEC-TDA chromatograms of the different γ-PGA samples: refractive index signals of the COM-PGA (red line), MET-PGA powder (blue line), as well as of the R1 (purple line), R2 (green line) and R3 (black line) fractions (**A**). Average Mw, poly-dispersity index (Mw/Mn), intrinsic viscosity (IV) and percentage representativity of each γ-PGA fraction analyzed by SEC-TDA (**B**).

**Figure 4 polymers-14-01190-f004:**
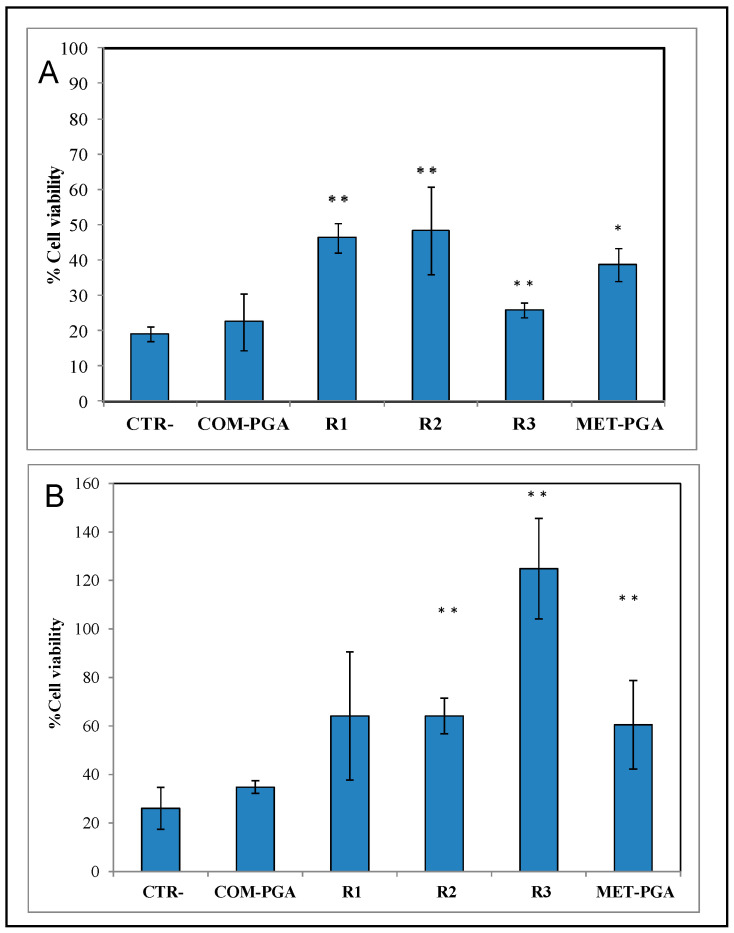
Dehydration assay (**A**) and oxidative stress assay (**B**) on HaCat cells as rescue effect of COM-PGA, MET-PGA and of γ-PGA-containing R1, R2 and R3 fractions, normalized over the positive control and compared to the negative control. (Statistical analysis: * *p* < 0.05 and ** *p* < 0.01 compared to the negative control).

**Figure 5 polymers-14-01190-f005:**
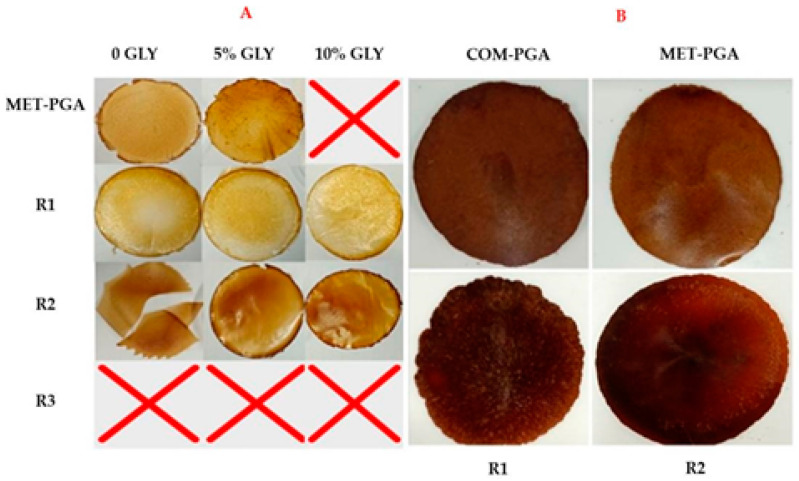
Films obtained by casting film forming solutions prepared with 800 mg of MET-PGA, R1, R2 and R3 γ-PGA fractions in the absence or presence of two different glycerol (GLY) concentrations; x, unhandleable sticky material (**A**). Films obtained by hot compression of 2 g of COM-PGA (γ-PGA), MET-PGA, R1 and R2 powders mixed with 5% GLY (**B**). Further experimental details are given in the text.

**Figure 6 polymers-14-01190-f006:**
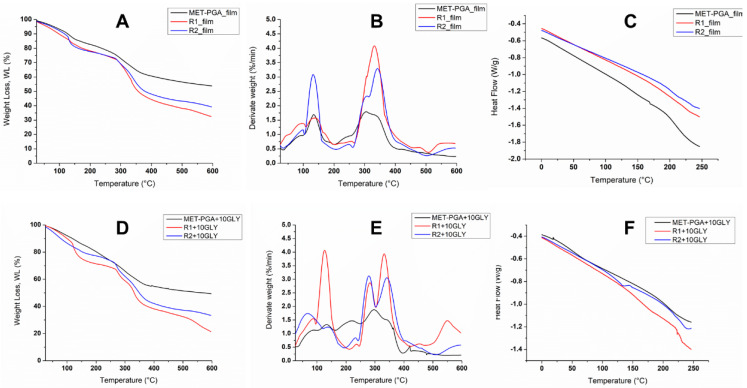
Thermogravimetric (**A**,**D**), differential thermogravimetric (**B**,**E**) and differential scanning calorimetric (second heating ramp) (**C**,**F**) thermograms of MET-PGA, R1 γ-PGA and R2 γ-PGA films, without (**A**–**C**) and with (**D**–**F**) 10% glycerol (10GLY).

**Table 1 polymers-14-01190-t001:** Summarized data of the γ-PGA filtration membrane-based purification process. (ND, not determined). Mass yield values were calculated as percentage ratio of the mass value of each sample on the mass value of the initial sample.

Sample	Volume (L)	pH	Conductivity (mS/cm)	Dry Weight (g)	Total γ-PGA (g)	% γ-PGA Recovery/Initial Sample	Mass Yield (%)
γ-PGA	10.0	6.2	14.0	272.5	140.3	-	-
Microfiltered γ-PGA	9.3	6.2	14.0	233.7	120.3	85.7	85.7
100 kDa Retentate (R1)	0.7	6.3	4.5	31.3	21.8	15.5	11.4
100 kDa Permeate (P1)	11.6	6.3	8.8	187.7	94.5	67.4	68.9
3 kDa Retentate (R2)	1.0	6.5	5.8	27.1	15.9	11.3	10.0
3 kDa Permeate (P2)	11.6	6.3	7.6	152.5	76.6	54.6	56.0
Nano-filtration Retentate (R3)	2.6	6.4	4.0	110.0	70.7	50.4	40.4
Nano-filtration Permetate (P3)	8.7	6.4	7.0	36.2	4.6	3.3	13.3

**Table 2 polymers-14-01190-t002:** Mechanical and hydrophilicity properties of the films obtained by either casting or hot compression of the different γ-PGA samples (ND, not detectable values because of the unmani-pulable (either * sticky or ** brittle) material obtained or because of *** the very low tensile strength (TS) values (<1.0 MPa) detected. Different small letters (a–g) indicate significant differences among the values reported in each column for each procedure (*p* < 0.05). Further experimental details are given in the text.

Procedure	Sample	GLY (%)	T (µm)	TS MPa	EB (%)	YM (MPa)	MC (%)	WS (%)
Casting/Drying	COM-PGA	0	ND *					
		5	ND *					
		10	ND *					
	MET-PGA	0	128 ± 4 ^a^	1.1 ± 0.1 ^a^	20 ± 3 ^a^	302 ± 20 ^a^	15 ± 1 ^a^	82 ± 1 ^a^
		5	143 ± 9 ^b^	<1.0	56 ± 5 ^b^	ND ***	17 ± 1 ^a^	84 ± 2 ^a^
		10	ND *					
	R1	0	149 ± 1 ^b^	7.3 ± 2.0 ^b^	0.5 ± 0.1 ^c^	1927 ± 176 ^b^	16 ± 1 ^a^	99 ± 1 ^b^
		5	178 ± 2 ^c^	6.1 ± 0.5 ^b^	13 ± 1 ^a^	678 ± 29 ^c^	16 ± 1 ^a^	99 ± 1 ^b^
		10	193 ± 4 ^d^	1.2 ± 0.4 ^a^	100 ± 27 ^d^	230 ± 80 ^a^	16 ± 1 ^a^	99 ± 1 ^b^
	R2	0	ND **					
		5	ND **					
		10	89 ± 1 ^d^	1.1 ± 0.2 ^a^	58 ± 1^b^	318 ± 45 ^a^	13 ± 1 ^a^	99 ± 1 ^b^
**Procedure**	**Sample**	**GLY (%)**	**T (µm)**	**TS MPa**	**EB (%)**	**YM (MPa)**	**MC (%)**	**S (%)**
Hot compression	COM-PGA	0	364 ± 61 ^a^	<1.0	8 ± 1 ^a^	ND ***	8 ± 1 ^a^	92 ± 2 ^a^
		5	421 ± 20 ^a^	<1.0	17 ± 3 ^b^	ND ***	11 ± 1 ^b^	96 ± 1 ^a^
		10	ND *					
	MET-PGA	0	ND **					
		5	472 ± 17 ^b^	<1.0	8 ± 1 ^a^	ND ***	7 ± 1 ^a^	91 ± 1 ^a^
		10	591 ± 17 ^c^	<1.0	10 ± 1 ^a^	ND ***	9 ± 1 ^ab^	99 ± 1 ^a^
	R1	0	499 ± 10 ^b^	<1.0	38 ± 2 ^c^	ND ***	11 ± 1 ^b^	62 ± 1 ^b^
		5	537 ± 32 ^bc^	<1.0	113 ± 6 ^d^	ND ***	11 ± 1 ^b^	71 ± 1 ^c^
		10	454 ± 10 ^ab^	<1.0	159 ± 8 ^e^	ND ***	11 ± 1 ^b^	99 ± 1 ^a^
	R2	0	463 ± 13 ^b^	<1.0	31 ± 9 ^c^	ND ***	12 ± 2 ^b^	99 ± 1 ^a^
		5	415 ± 10 ^a^	<1.0	77 ± 1 ^f^	ND ***	12 ± 2 ^b^	99 ± 1 ^a^
		10	430 ± 14 ^ab^	<1.0	785 ± 20 ^g^	ND ***	13 ± 2 ^b^	99 ± 1 ^a^

**Table 3 polymers-14-01190-t003:** Glass transition temperatures (Tg) and the starting endothermic peak (Tpeak) of all the films.

Samples	(DSC) Tg (°C)	(TGA) Tpeak (°C)
MET-PGA	125	300–350
MET-PGA+10%GLY	50	300–350
R1 γ-PGA	163	300–330
R1 γ-PGA +10%GLY	60	280–330
R2 γ-PGA	82	300–340
R2 γ-PGA +10%GLY	56	280–340

## Data Availability

The data that support the findings of this study are available on request from the corresponding author.

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
