# Peer review of "Exploiting Potential Biotechnological Applications of Poly-γ-glutamic Acid Low Molecular Weight Fractions Obtained by Membrane-Based Ultra-Filtration"

_polymers, 2022, doi:10.3390/polym14061190_

Round 1

Reviewer 1 Report

The article is focused on the potential of application of low molecular weight poly(gamma -glutamic acid) (PGA), especially in cosmetic sector. PGA is a promising biodegradable polymer from bacterial sources with intense applications in the field of medicine, wastewater treatment, food and cosmetics, … The originality of the articles lies in this study of the potential of this particular type of PGA for bioapplications eventhough the process has already been published including purification by ultrafiltration. It is curious that only one paper (ref. 6) is cited on this aspect of the process when they could cite this procedure as already known in the introduction section or at least compare in the discussion section their procedure with other recent studies (example given,“ An integrated strategy for recovery and purification of poly-γ-glutamic acid from fermentation broth and its techno-economic analysis” Zhang, X et al. Separation and purification technology , 2021, Vol.278, p.119575 DOI: 10.1016/j.seppur.2021.119575; and “Fermentative production of poly (γ-glutamic acid) from renewable carbon source and downstream purification through a continuous membrane-integrated hybrid process” Kumar, R; et al. Bioresource technology , 2015, Vol.177, p.141-148, 1873-2976; DOI:10.1016/j.biortech.2014.11.078 ). Nevertheless, the characterization of the fraction is detailed and somehow interesting even if the presentation would be improved and that the application potential is only assessed as a first approach. Furthermore, I would suggest the following amendments to the manuscript as outlined below

  1. The end of the penultimate paragraph of the introduction could be improved by adding some information about the purification process, including some of the information reported in the discussion section and relative to others groups work in that subject (see remarks above also).
  2. Line 191 and 227 the authors mention a standard but no information is given about it even in the material section. Especially regarding the average molar masses of this sample which are particularly interesting for the determination of dn/dc. Indeed, on this point, has this value also been evaluated with other real samples (i.e. R1, R2 and R3) with different molar masses, as large difference in molecular weight may affect the value (e.g. a high molecular weight compared to the lowest molecular weight).
  3. Line 391 page 10 the quality of Figure 2 must be improved.
  4. Line 396-454 and figure 12, it is difficult to relate the information given in table B and the RI chromatogram given in figure A. It would help even the expert if pic chosen for the value given in table B were positioned in figure A. Indeed, there is 3 molecular weight domains considered by the authors page 12 50to 70kDa; about 20kDa and 3-7kDa which could be positioned on the figure. In addition, if separation between low molecular weight and higher ones is clear, could give the authors give their methods to clear separate the several populations of higher molecular weight, in case of Com PGA and PET PGA by examples as it could only be done by deconvolution. If it is the case, could the authors discuss the pertinence of the value given in those case in table B (figure 2)?
  5. To complete previous remark and in correlation to the comments in the discussion (line 690-693), it is true that the molecular weight has been obtained by SEC-DTA but dual detection is sufficient for that and no information about the correlation with viscosity measurement is given. In addition, as PEG standard have also been used for calibration of the apparatus. Are molar masse sin correlation with universal calibration curves which could be obtained thank to that standard and the measurement of the viscosity values.
  6. Line 478-490, why the zeta potential evaluation to establish the stability of the solution is only done on com-PGA and not on some of the other fraction, especially the lowest molecular weight?
  7. It would be more interesting to combine figures 7, 8 and 9 to be able du compare the behaviour of the various PGA fraction on the same graph, and to use supporting information for some of the data if necessary. In addition, it would have been better to show only the TGA second run to facilitate de determination of the main transition on the thermogram.
  8. the article should be carefully proofread by the authors to avoid leaving some typos like “?-PGA” line 77; “costant” line 227, or the use of some abbreviation whose signification is not given FFF line 97 for the first time or the unfinished sentence on line 390.

Author Response

The article is focused on the potential of application of low molecular weight poly(gamma -glutamic acid) (PGA), especially in cosmetic sector. PGA is a promising biodegradable polymer from bacterial sources with intense applications in the field of medicine, wastewater treatment, food and cosmetics, … The originality of the articles lies in this study of the potential of this particular type of PGA for bioapplications eventhough the process has already been published including purification by ultrafiltration. It is curious that only one paper (ref. 6) is cited on this aspect of the process when they could cite this procedure as already known in the introduction section or at least compare in the discussion section their procedure with other recent studies (example given,“ An integrated strategy for recovery and purification of poly-γ-glutamic acid from fermentation broth and its techno-economic analysis” Zhang, X et al. Separation and purification technology , 2021, Vol.278, p.119575 DOI: 10.1016/j.seppur.2021.119575; and “Fermentative production of poly (γ-glutamic acid) from renewable carbon source and downstream purification through a continuous membrane-integrated hybrid process” Kumar, R; et al. Bioresource technology , 2015, Vol.177, p.141-148, 1873-2976; DOI:10.1016/j.biortech.2014.11.078 ). Nevertheless, the characterization of the fraction is detailed and somehow interesting even if the presentation would be improved and that the application potential is only assessed as a first approach. Furthermore, I would suggest the following amendments to the manuscript as outlined below

  1. The end of the penultimate paragraph of the introduction could be improved by adding some information about the purification process, including some of the information reported in the discussion section and relative to others groups work in that subject (see remarks above also).

-Thank you for the suggestion. New sentences were inserted in the introduction to deeper describe the PGA purification steps and recent literature papers were included in the references.

  1. Line 191 and 227 the authors mention a standard but no information is given about it even in the material section. Especially regarding the average molar masses of this sample which are particularly interesting for the determination of dn/dc. Indeed, on this point, has this value also been evaluated with other real samples (i.e. R1, R2 and R3) with different molar masses, as large difference in molecular weight may affect the value (e.g. a high molecular weight compared to the lowest molecular weight).

-We are not sure about which standard the reviewer is referring to. As reported in the material and method section at line 124-126 “All other chemicals used for the buffer of SEC-TDA and UHPLC analyses and the standards were of analytical grade and from Sigma-Aldrich (St. Louis, MO, USA)”, thus both L-glutamic acid standard and the γ-PGA standard were from Sigma-Aldrich.

-If the author refers to the averaged Mw of the standard of γ-PGA it was experimentally determined in our lab as 580 kDa ± 2, with an averaged polydispersity of 1.1 ± 0.1, as the Mw reported on the data sheet of the standard by SIGMA was > 750 kDa  and the range of polydispersity was 1.0 - 2.0.Thus these data were not precisely determined and besides they were obtained by MALLS, according to the manufacturer data sheet. We include the experimental data of the obtained averaged Mw and of the polydispersity of the γ-PGA standard in the manuscript, in the material and method section.

- In the material and method section, the formula to calculate the dn∙dc-1 was already reported (the formula now, in the revised version of the paper, is reported as #7), as described by the manufacturer of the SEC-TDA instrument (Viscotek, Malvern) at line 257.

                        [dn∙dc-1 = (Area RI/g·L-1·1000)/(kRI∙injection volume)∙RI of the solvent ……..(7)]

As visible from the formula, the determination of the dn∙dc-1 value is not a function of the Mw of the standard used to determine it but only of two experimental data: 1) the Area of the RI signal and 2) the concentration of the sample. In addition, the dn∙dc-1 value depends on the solvent used (the RI of the solvent) and on the injection volume. It depends also on the constant of the instrument, kRI (9.72∙107), that is obtained after calibration with an external standard that is the PEO. This external calibration allows to determine the dn∙dc-1 value independently from the Mw of the standard used (universal calibration). Thus, it is not necessary to calculate the dn∙dc-1 value of a specific macromolecule every time that its averaged Mw changes. This is one of the major advantages in using this instrument and this technology (SEC-TDA). For further technical information we suggest you to visit the manufacturer web site (Viscotek, information available from http://www.viscotek.com).

The methods paragraph was revised to improve the understanding of the technique.

  1. Line 391 page 10 the quality of Figure 2 must be improved.

-The quality of the image of Figure 2 has been improved. Thank you for the suggestion.

  1. Line 396-454 and figure 12, it is difficult to relate the information given in table B and the RI chromatogram given in figure A. It would help even the expert if pic chosen for the value given in table B were positioned in figure A. Indeed, there is 3 molecular weight domains considered by the authors page 12 50to 70kDa; about 20kDa and 3-7 kDa which could be positioned on the figure. In addition, if separation between low molecular weight and higher ones is clear, could give the authors give their methods to clear separate the several populations of higher molecular weight, in case of Com PGA and PET PGA by examples as it could only be done by deconvolution. If it is the case, could the authors discuss the pertinence of the value given in those case in table B (figure 2)?

-A supplementary material, Figure S2, was inserted to better comply to the referee request, to show the complete signals output and to highlight the specific integration (with the dashed lines) of the SEC-TDA analyses of the samples at level of the inflection points that were selected in relation to light scattering (LALS/RALS) signals and the intrinsic viscosity signal, and that allowed to highlight differences in the polymeric populations (families).

  1. To complete previous remark and in correlation to the comments in the discussion (line 690-693), it is true that the molecular weight has been obtained by SEC-DTA but dual detection is sufficient for that and no information about the correlation with viscosity measurement is given. In addition, as PEG standard have also been used for calibration of the apparatus. Are molar masse sin correlation with universal calibration curves which could be obtained thank to that standard and the measurement of the viscosity values.

-The reviewer is right. As mentioned above the standard with which the SEC-TDA instrument is calibrated is a polyethylene oxide (PEO) standard (22 kDa PolyCAL, Viscotek, Malvern, Italy) and a universal calibration is performed. The average Mw, the polidispersity index (Mw/Mn) and the intrinsic viscosity (IV) obtained by the SEC-TDA analyses were derived by solving the following equations that are all related to each others:

RI signal = kRI · dn∙dc-1 · C;

   VIS signal = kVIS · IV · C;

   LALS signal = kLALS · Mw · (dn∙dc-1 )2 ·C;

where IV is the intrinsic viscosity (dl·g-1); C is the concentration (mg·ml-1); dn∙dc-1 is the refractive index increment (ml/g) and kRI, kVIS, and kLALS are instrumental constants obtained by the universal calibration with PEO, performed according to the manufacturer procedure (Viscotek, information available from http://www.viscotek.com). This explanation and the three equations mentioned above have been added in the material and method section of the paper to be clearer. Thank you for the suggestion.

  1. Line 478-490, why the zeta potential evaluation to establish the stability of the solution is only done on com-PGA and not on some of the other fraction, especially the lowest molecular weight?

   -In  the revised version of the manuscript the zeta potential of MET PGA sample has been removed in order to avoid any type of misunderstanding.

  1. It would be more interesting to combine figures 7, 8 and 9 to be able du compare the behaviour of the various PGA fraction on the same graph, and to use supporting information for some of the data if necessary. In addition, it would have been better to show only the TGA second run to facilitate de determination of the main transition on the thermogram.

-Thank you very much for your valuable observations. TGA thermograms have been wholly revised and combined in order to both improve readability and to enhance their discussion. In addition, only the second heating run of DSC analysis have been reported and overlapped, as requested. A deeper explaination of the results have been provided. Thank you very much for giving us the opportunity to improve the paper with your proper suggestions.

  1. the article should be carefully proofread by the authors to avoid leaving some typos like “?-PGA” line 77; “costant” line 227, or the use of some abbreviation whose signification is not given FFF line 97 for the first time or the unfinished sentence on line 390.

-The manuscript was carefully revised to correct all the typing and grammar mistakes. Thank you.

Reviewer 2 Report

The article submitted for review concerns research on the biotechnological potential of poly-γ-glutamic acid. The discussed topic is important from the application point of view. In my opinion, the parts Introduction and Materials and Methods are prepared correctly and do not require improvement. However, the Results, Discussion and Conclusions should be improved.

In the Results section, the Authors provide an extensive but qualitative description of the obtained results, which are presented in the form of tables and graphs. Up to 14 pages were devoted to describe them. While the Discussion of the obtained results took only 2 pages.

I believe that some descriptions in Results can be shortened, because there is no need to duplicate the results which are visible in tables or charts. On the contrary Discussion should definitely be broadened. In its current form, it is too general. The same applies to the Conclusions.

In my opinion the description of the thermal analysis results raises some doubts. On the basis of the presented results, it is not possible to determine if water released in the first stage (up to 200 degrees centigrade) is only product or any other products are released. Moreover, the relatively high upper limit (up to 200 degrees centigrade) may suggest, the water that is released comes not only from the coordination zone but also from the condensation processes taking place between the functional groups of the polymers.

In line 657 the Authors probably considered the process of water evaporation from the sample, and not the melting of water.

I also would like to point out, the inconsistent interpretation of the DSC results. The Authors wrote that in the second heating cycle, the decrease in the thermal curve is connected with initial stage of melting the polymer. In general, the melting process (especially for purified compounds) on the DSC curve takes the form of a more or less sharp peak over at certain temperature range. In this case, a decrease is observed over a wide range of temperature (100-250 deg) but despite this, it is not possible to determine even the onset for this process. Additionally, it is enough to look at the TG / DTG curves to see that the tested samples above 200 deg. yield decomposition, what the authors wrote in line 599.

In this respect, the results of the TG / DTG and DSC analysis must be absolutely consistent, therefore the discussed description requires correction. Interpretation of TG / DTG / DSC data without evolving gas analysis (EGA) should be done with a great care.

Author Response

The article submitted for review concerns research on the biotechnological potential of poly-γ-glutamic acid. The discussed topic is important from the application point of view. In my opinion, the parts Introduction and Materials and Methods are prepared correctly and do not require improvement. However, the Results, Discussion and Conclusions should be improved.

In the Results section, the Authors provide an extensive but qualitative description of the obtained results, which are presented in the form of tables and graphs. Up to 14 pages were devoted to describe them. While the Discussion of the obtained results took only 2 pages.

I believe that some descriptions in Results can be shortened, because there is no need to duplicate the results which are visible in tables or charts. On the contrary Discussion should definitely be broadened. In its current form, it is too general. The same applies to the Conclusions.

-Thank you for the suggestion. The result section was reduced trying to avoid description of the processes and analyses reported in materials and methods, and not to duplicate results respect to the ones reported in figures and tables. Finally, the discussion was improved.

In my opinion the description of the thermal analysis results raises some doubts. On the basis of the presented results, it is not possible to determine if water released in the first stage (up to 200 degrees centigrade) is only product or any other products are released. Moreover, the relatively high upper limit (up to 200 degrees centigrade) may suggest, the water that is released comes not only from the coordination zone but also from the condensation processes taking place between the functional groups of the polymers.

-Thank you very much for your valuable observations. Actually, as widely reported in literature, the investigated region from room temperature up to about 150-200°C, can be ascribed to free, bound and strongly bound water releasing, mostly if hydrophilic and hygroscopic polymers are concerned. It should be considered that PGA based samples are characterized by the presence of many polar groups, as amino, carboxyl and hydroxyl residues, mostly involved in hydrogen bonding with water and, in any case, able to absorb and/or entrap water molecules in their three-dimensional network. It’s not a case that in a previous paper, some of the authors of the current paper demonstrated the same results for PGA based samples (Sabbah et al., 2020.  Polymers 12, 1613-1626); also most of the literature data related to PGA based samples agreed with this hypothesis: (Pereira et al., 2017. Carbohydrate Polymers, 157, 1862–1873; Pradeepkumar et al., Chemistry Select 2019, 4, 10225– 10235; Portilla-Arias et al., 2007, Polymer Degradation and Stability, 92, 1916-1924; PigÅ‚owska et al., 2020, Polymers 12, 357-371).

In line 657 the Authors probably considered the process of water evaporation from the sample, and not the melting of water.

-The referee observation is absolutely right. The line 657 refers to water evaporation process, associated to the endothermic peak. The expression was modified according the suggestion, thank you.

I also would like to point out, the inconsistent interpretation of the DSC results. The Authors wrote that in the second heating cycle, the decrease in the thermal curve is connected with initial stage of melting the polymer. In general, the melting process (especially for purified compounds) on the DSC curve takes the form of a more or less sharp peak over at certain temperature range. In this case, a decrease is observed over a wide range of temperature (100-250 deg) but despite this, it is not possible to determine even the onset for this process. Additionally, it is enough to look at the TG / DTG curves to see that the tested samples above 200 deg. yield decomposition, what the authors wrote in line 599.

-Dear reviewer, thank you for your relevant observation since you gave us the opportunity to reconsider this part of the paper by improving the resulting evaluable data. Firstly, as you properly suggested, the hypothesis related to melting phenomena associated with the starting endothermic peaks was removed from the manuscript. Actually, by performing DTA analysis, associated with TGA/DTG instrument software, it was possible to assess that, in the range of 25-600°C, no thermal transitions correlated to melting phenomena were found, as the only detected peaks corresponded to the ones of polymer thermal degradation. Since another referee suggested to drastically reduce the number of figures, DTA curves, overlapped with DTG curves of all samples, were reported in supplementary material, as for your check. Nevertheless, by DSC second heating run, it was possible to detect the glass transition temperatures highlighting a general plasticization effect of glycerol; the detected values have been introduced in Table 5 in the text together with the Tpeak coming from DTG curves. On behalf of the co-authors, I really thank you for your great, deep and remarkable support.

In this respect, the results of the TG / DTG and DSC analysis must be absolutely consistent, therefore the discussed description requires correction. Interpretation of TG / DTG / DSC data without evolving gas analysis (EGA) should be done with a great care.

-Once again, the referee observation is correct. Actually, EGA analysis were not performed since no one of the authors had a TGA_MS equipment. Anyway, we are thinking to go in depth with further investigations on this topic and your worthy suggestion will be taken in consideration for the future activity. Anyway, we really thank you since your whole demands and suggestions helped us to improve the paper.

Reviewer 3 Report

List of comments/recommendations are presented below:

  1. What was the initial value of feed transmembrane pressure in the UF and NF steps of γ-PGA purification? What did determine the choice of this value?
  2. Did the authors carry out any washing procedure for the membranes in different UF and NF steps to deal with membrane fouling phenomenon? This could help to prevent flux decrease during the filtration process. Moreover, a higher amount of R1, R2 and R3 retentates could be obtained as the end-product and, consequently, a higher amount of γ-PGA could be recovered.
  3. What was the reason for such a non-linear of TMP curve trend in the DF phase of the 100 kDa membrane process (Figure 3A)?
  4. What is the concentration factor and how was it calculated (lines 337, 404, 411)? Is it similar to the membrane selectivity value?
  5. What are the values for the membrane selectivity in each filtration steps (calculated according to equation 2)?
  6. The text in lines 666-715 seems to be more suitable for the Introduction part, but not for Discussion. Discussion could be supplemented with more detailed description of the results (e.g., which of studied COM-PGA, MET-PGA, R1,R2 and R3 containing γ-PGA is considered to have the most optimal properties regarding biological activity; which of the procedures is preferable for the preparation of γ-PGA films, etc.).

Author Response

List of comments/recommendations are presented below:

-What was the initial value of feed transmembrane pressure in the UF and NF steps of γ-PGA purification? What did determine the choice of this value?

-The initial values of transmembrane pressure are 0.22 bar for 100 kDa membrane process,  0.20 bar for 3 kDa membrane process and 11.8 bar for the nano-filtration membrane process, as clearly visible in Figure 2 A, B and C. It depends on the membrane cut-off, on the ultrafiltration system speed of the pump and on the type of membrane cassette used, as reported in material and method.

-Did the authors carry out any washing procedure for the membranes in different UF and NF steps to deal with membrane fouling phenomenon? This could help to prevent flux decrease during the filtration process. Moreover, a higher amount of R1, R2 and R3 retentates could be obtained as the end-product and, consequently, a higher amount of γ-PGA could be recovered.

-The first UF/DF phase was run on an automatic tangential flow filtration system (Uniflux 10, UNICORN, GE Healthcare, USA), that is automatically increasing TMP to sustain flux. The cassettes used, and the system itself were not suitable for backflushing runs to reduce fouling. However, in this case, having separated in advance, through microfiltration, the solid, the mechanism of concentration polarization typical of UF system may be considered the sole responsible for flux decrease. On the other hand, after diafiltration and collection of the retentate sample washes of the membrane were performed for each purification step, and the volume of the washes were added to the retentate ones, and the final volume is reported in the paper as retentate sample.

-What was the reason for such a non-linear of TMP curve trend in the DF phase of the 100 kDa membrane process (Figure 3A)?

-During the diafiltration (DF) step a volume of deionized water was added but then the sample was also further concentrated. The increase of TMP and the flux decrease are explained as during the DF also a simultaneous concentration of the samples occurred and the final volume was lower than the one obtained during ultrafiltration step.

-What is the concentration factor and how was it calculated (lines 337, 404, 411)? Is it similar to the membrane selectivity value?

-The concentration factors are the ratio of the initial volume of a sample before ultrafiltration on the final volume of the same sample as retentate, calculated for each step of membrane purification. For example for the 100 kDa membrane concentration the calculation is 9.3 L (initial volume obtained after microfiltration)/0.7 L (final retentate volume) = 13.3. It is a very simple and well known approach thus we do not think we have to insert it as formula in the manuscript.

-What are the values for the membrane selectivity in each filtration steps (calculated according to equation 2)?

-The reviewer is right, the selectivity values are missing. They have been inserted in the text in the result sections.

-The text in lines 666-715 seems to be more suitable for the Introduction part, but not for Discussion. Discussion could be supplemented with more detailed description of the results (e.g., which of studied COM-PGA, MET-PGA, R1,R2 and R3 containing γ-PGA is considered to have the most optimal properties regarding biological activity; which of the procedures is preferable for the preparation of γ-PGA films, etc.).

-The referee observation was absolutely pertinent. This part of the paper was extensively revised. In particular in the final part of the discussion some considerations on the preferable method for the preparation of γ-PGA films and on the biological activity have been added. Thank you very much for your worthy suggestion.

Reviewer 4 Report

This work investigated precipitation and UF/NF purification methods to fractionate gamma-PGA fractions. The resulting low MW PGA fraction was fabricated into films using different concentrations of glycerol as plasticizer, then thoroughly characterized. The work is interesting, thorough, and backed by sufficient data. Some concerns remain, though, hence the reviewer recommends publication after taking the following comments into account.

1) All figures are in different formats, please unify them using professional software such as ORIGIN and SigmaPlot. If unavailable, at least plot them using the same format in excel.

2) In line 48, the authors mean alpha-helix and beta-sheet confirmation?

3) Why did the authors diafilter the retentate after each filtration experiment? To extract the remaining permeating species from the retentate? Then, the permeate during the diafiltration was mixed with the first batch of permeate? What was the mass yield in each step?

4) I do not quite follow the diafiltration step after UF. The authors added two volumes of DI water to dilute the retentate, and why did the TMP increase and the flux decreased? Dilution would have alleviated the osmotic pressure and enhanced the flux.

5) It would be good to re-emphasize why fractionation is useful compared to using crude batch.

Author Response

1) All figures are in different formats, please unify them using professional software such as ORIGIN and SigmaPlot. If unavailable, at least plot them using the same format in excel.

-The figures were revised. Thank you.

2) In line 48, the authors mean alpha-helix and beta-sheet confirmation?

-Yes, thank you for underlying the typing mistakes. We corrected them.

3) Why did the authors diafilter the retentate after each filtration experiment? To extract the remaining permeating species from the retentate? Then, the permeate during the diafiltration was mixed with the first batch of permeate? What was the mass yield in each step?

-Dia-filtration steps were performed in order to reduce the retention on the membranes of undesired molecules or of low molecular weight γ-PGA that could remain stacked on them. Molecules having a molecular weight similar or lower the cut-off of the membranes could even be retained due to the well-known phenomenon of concentration polarization with a formation of a gel-layer on the UF membranes themselves. The permeates of each dia-filtration step were then mixed with the permeate obtained during the ultra-filtration step. The mass yield values of each step have been added to Table 1, as requested by the reviewer. Thank you for the suggestion.

4) I do not quite follow the diafiltration step after UF. The authors added two volumes of DI water to dilute the retentate, and why did the TMP increase and the flux decreased? Dilution would have alleviated the osmotic pressure and enhanced the flux.

-During the dia-filtration step the volume of DI water was added in more than one time in order to avoid extremely dilution of the retentate and, in any case, the final volume of the retentate after dia-filtration was lower than the one obtained during ultrafiltration step as the sample was further concentrated. As during the DF also a simultaneous concentration of the samples occurred, the increase of TMP and the flux decrease are explained.

5) It would be good to re-emphasize why fractionation is useful compared to using crude batch.

-Thank you for your suggestion. To better explain the necessity of a fractionation we changed a part of the introduction section, on the other hand we think that this point is very well emphasize in the discussion in the lines 729-744.

Round 2

Reviewer 3 Report

The Authors have made the necessary changes and corrections to the manuscript. it can be recommended for publication.